Identification of rare alternative splicing events in MS/MS data reveals a significant fraction of alternative translation initiation sites

Kroll José E. 1 2
de Souza Sandro J. 2
de Souza Gustavo A. 3 g.a.d.souza@medisin.uio.no
1 Institute of Bioinformatics and Biotechnology , Natal , Brazil
2 Brain Institute, UFRN , Natal , Brazil
3 Department of Immunology and Centre for Immune Regulation, Oslo University Hospital HF Rikshospitalet, University of Oslo , Oslo , Norway
Emes Richard
Electronic publication date: 2014 Nov 13
Publication date: 2014
Volume: 2
Electronic Location ID: e673
Received 2014 Oct 1; Accepted 2014 Oct 30
Copyright: © 2014 Kroll et al.
Copyright year: 2014
Copyright holder: Kroll et al.
License: This is an open access article distributed under the terms of the Creative Commons Attribution License, which permits unrestricted use, distribution, reproduction and adaptation in any medium and for any purpose provided that it is properly attributed. For attribution, the original author(s), title, publication source (PeerJ) and either DOI or URL of the article must be cited.
License URL: https://creativecommons.org/licenses/by/4.0/

Keywords: Mass spectrometry, Proteomics, Alternative splicing events, Peptide identification, Translation initiation sites

Funding: CNPq 501891/2013-7 CNPq 483775/2012-6 CAPES edital 051/2013 CNPq 400392/2014-3 UiO Norwegian South-East Health Authority JEK is supported by a post-doctoral fellowship from CNPq (501891/2013-7). This research was supported by grants from CNPq (483775/2012-6) and CAPES (edital 051/2013), both to SJS. GAdS is a Special Visiting Scientist to the Brain Institute – UFRN (supported by a CNPq grant 400392/2014-3 to SJS). GAdS and the Proteomics Core Facility are supported by grants from UiO and the Norwegian South-East Health Authority (Helse Sør-Øst). The funders had no role in study design, data collection and analysis, decision to publish, or preparation of the manuscript.

==============================
Integration of transcriptome data is a crucial step for the identification of rare protein variants in mass-spectrometry (MS) data with important consequences for all branches of biotechnology research. Here, we used Splooce, a database of splicing variants recently developed by us, to search MS data derived from a variety of human tumor cell lines. More than 800 new protein variants were identified whose corresponding MS spectra were specific to protein entries from Splooce. Although the types of splicing variants (exon skipping, alternative splice sites and intron retention) were found at the same frequency as in the transcriptome, we observed a large variety of modifications at the protein level induced by alternative splicing events. Surprisingly, we found that 40% of all protein modifications induced by alternative splicing led to the use of alternative translation initiation sites. Other modifications include frameshifts in the open reading frame and inclusion or deletion of peptide sequences. To make the dataset generated here available to the community in a more effective form, the Splooce portal (http://www.bioinformatics-brazil.org/splooce) was modified to report the alternative splicing events supported by MS data.

Introduction

The development of large-scale technologies, including genomics, has revolutionized life sciences. For example, the sequencing of the human genome in 2001 was a milestone in the characterization of our genetic framework (Lander et al., 2001; Venter et al., 2001). The advancement of sequencing technologies in the last few years has allowed the genome sequencing of more than a thousand human individuals (1000 Genomes Project) (The 1000 Genomes Project Consortium, 2012). Likewise, the characterization of the transcriptome was also facilitated by these new sequencing technologies. RNA-Seq techniques have allowed the identification of transcripts with low copy numbers. Thus, the complete characterization of the transcriptome of different cell types is already a reality today (Au et al., 2013; Peng et al., 2012; Xue et al., 2014). We know for example about the large variability found in the transcriptomes of eukaryotes due to alternative splicing and alternative polyadenylation. As a consequence of the emergence of these technologies, an explosion of this type of data in public databanks and data repositories is already occurring and exponential growth is expected for the next years. Improving bioinformatics capabilities is crucial for the processing, storage and interpretation of results from large-scale technologies.

While the technologies for sequencing of nucleic acids developed at an impressive speed, the same did not happen with technologies for sequencing amino acids and proteins. Recently, mass spectrometry-based proteomics achieved enough comprehensiveness and throughput to allow in-depth characterization of “complete proteomes” (Beck et al., 2011; Nagaraj et al., 2011). However, proteomic data acquisition is still restricted to few groups, even though public availability of high depth proteomic data is increasing (Desiere et al., 2006; Perez-Riverol et al., in press; Vizcaino et al., 2013; Vizcaino et al., 2014).

Alternative splicing is defined, basically, as a process in which identical pre-mRNA molecules are processed in different ways in terms of usage of splice sites. It is a fundamental process in all multi-cellular organisms being responsible for the creation of a large diversity of proteins from a relatively small number of genes (Cork, Lennard & Tyson-Capper, 2012). Alternative splicing events (ASE) have been extensively characterized using transcriptome data. On the other hand, only recently proteome data have been used for global discovery of ASEs (Brosch et al., 2011; Severing, Van Dijk & Van Ham, 2011; Tress et al., 2008). This can be explained by two factors: first, limitations in data acquisition, such as the lower dynamic range of rarer isoforms and consequently its difficulty in collecting good quality fragmentation spectrum, resulting in poorer scoring and higher chances for false-discovery reporting; second, protein identification by mass spectrometry is still routinely performed through the use of protein databases cataloged and curated by public repositories such as nrNCBI and Uniprot. Most of these databanks contain only a limited number of protein sequence isoforms, and single nucleotide polymorphisms and ASEs are normally under-represented. This is generally so because peptide identification approaches in proteomics mostly use probabilistic-based algorithms, and excessively large databases would result in spurious spectral matches and, therefore, reduced number of positive identifications (Perez-Riverol et al., 2011; Wang et al., 2012; Woo et al., 2014). Thus, new approaches should be developed where ASEs can be investigated without compromising database size and protein identification rates. Several researchers have created strategies that use MS data repositories such as Peptide Atlas and in silico protein database design using nucleotide sequence repositories or merging protein sequence databases (Blakeley et al., 2010; Brosch et al., 2011). However, very few had applied RNA-Seq data to offer isoform information at the transcriptome level, which then could be validated at the protein level. For example, Sheynkman and colleagues (2013) developed a strategy where RNA-Seq and MS data collected from the same samples had been applied for the identification of splice junction peptides. However, applying such different expertise in any project might not be a reality for a majority of laboratories. Therefore, creating strategies that rely on heavy bioinformatics analysis of nucleotide de novo sequence and validation through MS is relevant.

Here, we investigated whether ASEs could be satisfactorily identified using size-limited FASTA database, built from repositories of expressed sequences, which was then challenged by MS data. Our group had recently developed Splooce, a database that integrates information from transcriptome analysis, including RNA-Seq, to identify splicing variants (Kroll et al., 2012). Protein entries created from Splooce were evaluated using MS/MS analysis, and a large number of novel proteins isoforms were identified. Surprisingly we found that around 40% of all modifications at the protein level were related to the use of alternative translation initiation sites (TIS).

Materials & Methods

Protein variants identification using mass spectrometry and MaxQuant

Predicted proteins were collected from the Splooce website. Since Splooce does not provide FASTA files and due to the complexity of our needs (large scale analysis), a robot-type script was developed to query alternative splicing events and their specific data, such as predicted proteins. Entries showing alternative splicing events supported only by ESTs and/or RNASeq expressed sequences were selected. Those events were tagged as rare since they were not found in the set of full-insert cDNA sequences (RefSeq, mRNA), which usually have well characterized coding sequences. Any pattern of combined alternative splicing event was allowed. As a default parameter, Splooce only reports events that are supported by at least two expressed sequences. For the prediction of protein sequences, Splooce uses a simple ab-initio strategy. Briefly, human entries from the Reference Sequence database (Pruitt et al., 2014) were modified by introducing alternative splicing patterns observed from the transcriptome data. Thus, full-length alternative cDNA sequences were created from expressed sequence fragments that often cover only a small fraction of coding sequences. As a final step, prior to the translation process, new open reading frames are predicted based on their length (largest one). Only alternative predicted proteins showing alterations on their amino acids composition were selected and, prior to be stored in the FASTA file, the sequences were tagged following the rule: REFSEQ_NAME# (EVENT_TYPE:SPLOOCE_ID). Our final set of predicted proteins, containing 120,299 entries, can be downloaded from http://www.bioinformatics-brazil.org/~jkroll/sploocemm. Additionally, we developed a simple tracking tool, available in http://www.bioinformatics-brazil.org/cgi-jkroll/msretry.pl, which users can use to easily recover information from any entry stored in the provided FASTA file. Human entries from Uniprot (Reference Proteome, including 89,628 canonical and isoform entries, downloaded 16th Dec 2013 from http://www.uniprot.org) (Magrane & UniProt Consortium, 2011) were added to the Splooce database to facilitate the visualization of identified peptides that are not unique to the Splooce set. Original identifiers from Uniprot were maintained throughout all further analyses. The final database contained 209,927 entries (89,628 and 120,299 from Uniprot and Splooce, respectively).

We submitted the collection of entries from Splooce plus Uniprot to a dataset of MS/MS peptide information collected from 11 tumor cell lines that were publicly available at the Tranche Network (currently discontinued (Perez-Riverol et al., in press)). The whole collection of MS data was derived from the laboratory of Dr. Mathias Mann (Geiger et al., 2012). Four RAW files from this dataset were not used because they were apparently corrupted in the depository. We submitted the remaining files to a MaxQuant (version 1.4.1.2) (Cox & Mann, 2008) search using the following parameters: trypsin with no proline restriction as enzyme, initial search with a precursor mass tolerance of 20 ppm that were used for mass recalibration; main search precursor mass and fragment mass were searched with mass tolerance of 6 ppm. The search included variable modifications such as Met oxidation, N-terminal acetylation (protein), and Pyro-Glu (Q)(E). Carbamidomethyl cysteine was added as a fixed modification. Minimal peptide length was set to 7 amino acids and a maximum of two miscleavages were allowed. The false discovery rate (FDR) was set to 0.01 for peptide and protein identifications. In the case of identified peptides that are shared between two proteins, these are combined and reported as one protein group. Protein table output was filtered to eliminate the identifications from the reverse database, and common contaminants.

Protein variants identification using a de novo strategy

We also decided to test the ability to identify peptides characterizing ASEs using a de novo approach rather than a probabilistic one using a database. MS raw files were submitted to de novo sequence identification using the PEAKS software (Ma et al., 2003). Parameters were set as: (i) trypsin with no proline restriction as enzyme, (ii) two miscleavages allowed and (iii) precursor ion and fragment ion error of 10 ppm. Furthermore, carbamidomethyl (Cys) as fixed modification, while protein N-term acetylation, Met oxidation and pyro-Glu (Q/E) were also allowed as variable modifications. Only peptide sequences with more than 80% average coverage certainty were selected for further analysis. Coverage certainty is calculated on an amino acid per amino acid basis, i.e., only in cases where the software was able to precisely detect mass of the amino acid removed from two neighboring daughter ions.

Identification of peptides supporting alternative splicing events

The output file of identified peptides obtained from MaxQuant and PEAKS were filtered for peptides observed uniquely on Splooce entries. As described above, all MaxQuant peptides showing reversed and contaminant tags were removed from the data set. The resulting peptides were then compared against an unmodified set of RefSeq sequences, which Splooce uses as template for predicting new proteins. Any peptide observed for a Splooce entry, but not observed for its respective unmodified RefSeq, was classified as an ASE supporting peptide since it aligns uniquely to the alternative protein sequence. Additionally, any ASE supporting peptides matching the beginning of proteins were classified as alternative translation start sites.

A clear limitation in a “database-based” approach is a reduction in peptide/protein identification due to an increase in the search space by creating an excessively large database. Therefore we restricted our database to a size approximately twice as big as Uniprot. Protein identification using our database obtained approximately 500 proteins less than the original publication, a variation of less than 5%. Since the original publication used a version of the discontinued International Protein Index database, we also submitted the dataset to Uniprot database without our in house Splooce sequences (data not shown), since Uniprot and IPI would have closer number of entries and therefore, similar search space (Griss et al., 2011). The Uniprot result identified approximately 200 proteins less than the original publication. Such differences are probably due to: (i) different identified unique entries in Uniprot or IPI, (ii) small differences in the parameters between our MaxQuant search and the original publication, and/or (iii) differences in MaxQuant performance since we used an updated version compared to the one used the original publication. Regardless, we concluded that even doubling the database size with Splooce entries, protein identification penalty was irrelevant for the approach efficiency.

Results and Discussion

Identification of splicing variants in the MS/MS data

Splooce was used as a source to create a database of predicted protein isoforms in FASTA format, which was then searched against MS/MS spectra. A data set of 120,299 non-redundant protein sequences was created based on rare ASEs that were not observed for full-insert cDNA sequences (see Experimental Procedures for more details). That data set was merged to 89,602 Uniprot entries from the December 2013 release. A public collection of MS RAW files was then selected for protein identification. Only files from a publication that reported good level of instrument sensitivity and proteomic depth (Geiger et al., 2012) were used and the MS dataset was challenged against the Splooce-derived protein sequences using two peptide identification approaches, one based in probabilistic method and another one based on de novo sequencing (Fig. 1). Both methods offer unique advantages and limitations. De novo sequencing provides unbiased peptide identification, not limited to its theoretical existence in a database. On the other hand, sequence information can only be obtained from good to high quality MS/MS data, and partial sequence information is generally discarded. Algorithms using a protein database overall offer a higher identification rate, since partial sequence information, together with accurate mass measurement of the precursor peptide ion, can still provide positive identification. De novo data also offer additional possibilities since once a given sequence information is obtained it can be aligned against sequence repositories to provide protein identification.

Figure 1 Experimental design flowchart.

Briefly, public MS data from 11 cell lines (Geiger et al., 2012) were submitted to peptide identification using a Splooce database either by a probabilistic approach (MaxQuant) or a de novo approach (PEAKS). Identified peptides were sorted and those characterizing alternative splicing events not present in Uniprot were compared.

Initial analysis using the probabilistic approach (MaxQuant) allowed us to identify a total of 142,926 unique peptides representing 11,237 protein groups. File S1 reports the MaxQuant peptide output containing the identification features for both the total peptides identified and the ones identified only in the Splooce database. As expected, the vast majority (142,008) of these peptides are already present in Uniprot. However, 911 peptides, representing 808 ASE, were only observed for Splooce entries.

We next plotted individual peptide intensities and scores from both the complete peptide dataset and peptides uniquely identified in Splooce. Data overview of the complete dataset showed, as previously reported, an intensity span of 7 orders of magnitude. The peptides characterizing the rare ASEs were observed mostly at the bottom half of the intensity distributions, with an average distribution approximately one order of magnitude lower than the complete Uniprot set (Fig. 2A). While the score distribution seemed similar, ASE-derived peptides, on average, had a lower distribution (Fig. 2B), which could be a consequence of poorer MS/MS from lower intensity ions.

Figure 2 Intensity and scoring distribution for all identified peptides.

Peptide signal intensity (log10) (A) and scoring (B) distribution for all peptides (ALL) and sorted alternative splicing events (ASE) in the probabilistic approach. ASE peptides were on average close to an order of magnitude less abundant than the whole peptide population, consequently with lower average scoring.

In addition, the same RAW files collection was submitted to PEAKS, a software capable of determining a MS/MS sequence without the support of a database. Since no FDR can be estimated without the support of reversed sequences artificially created from a database, this analysis was restricted to spectra where fragment ion mass sequences could be measure with an average confidence of at least 80%. Using this approach, approximately 50,000 peptides were identified in Uniprot and Splooce (data not shown), and from those only 236 peptides, confirming 218 splicing events, could be identified in the same Splooce-derived database as used in the probabilistic approach. From those, 134 ASE were already observed in the probabilistic approach. By merging the results of the two strategies, we characterized a total of 892 ASE (Files S2 and S3). However, it is important to note that, while both de novo and probabilistic methods have their own FDR calculations, we did not perform any additional validation to avoid error propagation from merging results from the two different approaches. Our objective here was mostly to investigate the most efficient method based on the reported findings.

As expected, the de novo method identified a smaller proportion of proteins and peptides than the probabilistic method when submitted to a BLAST-like alignment versus the same Splooce database. In fact, a smaller number of splicing events were detected in the de novo method when compared to the probabilistic one. An explanation for this could be that since most ASE events characterized by the probabilistic method are seen in the bottom part of signal intensity, they most probably generated partial MS/MS information that did not fulfill the criteria required by us for reporting good quality de novo sequences. With this observation we therefore conclude that performing a probabilistic method using an in house database generates more information than de novo sequencing.

The frequency of each type of alternative splicing was next calculated for all events identified in our strategy. Simple events like exon skipping, alternative splice borders and intron retention corresponded to 463 of the total number of events identified by MS/MS data and showed proportional frequencies when compared to general Splooce statistics (Table 1). Moreover, no ASEs resulting from dual-specificity splice sites were identified, since these events are very uncommon and usually found within UTR sequences (Zhang et al., 2007). Splooce is also a database that focus on the analysis of combined ASEs (CASEs), and it was previously shown that approximately half of all alternative expressed sequences may have more than one ASE along their sequences (Kroll et al., 2012). The analysis presented here confirms the same finding at the proteome level. Among the total amount of events identified by MS/MS data, 429 were classified as complex. The most frequent combined event was the skipping of several adjacent exons (up to 11 exons), followed by adjacent alternative splice sites.

Table 1 Amount of simple alternative splicing events identified by the MS/MS analysis compared to the total number of corresponding events available from the Splooce database.

Alternative splicing event	Total events from Splooce	Events identified by
the MS/MS analysis	
Exon skipping	38,060 (35%)	182 (39%)	
Alternative 3’ splice site	30,172 (29%)	130 (28%)	
Alternative 5’ splice site	27,585 (25%)	90 (20%)	
Intron retention	12,632 (11%)	61 (13%)	
Dual-specific splice site	112 (0%)	0 (0%)	

Alternative TIS represents the majority of events at the proteome level

We further explored what types of events were observed in the identified peptides. Interestingly, 355 ASEs, out of the 892 (40%), showed a pattern consistent with the use of an alternative TIS due to an ASE (Fig. 3, File S2). The remaining 537 proteins showed different types of variations along their protein sequences (File S3). Files S2 and S3 not only contain a resumed version of the results described in this section, but also report protein sequence alignments for Uniprot and Splooce sequences of all proteins identified with a rare ASE. Peptides shared between both databases, in addition to the Splooce-specific peptide(s), are highlighted in the alignment. Most importantly, each alignment contains a link to the Splooce website where information and statistics for that rare ASE can be collected.

Figure 3 Alignments between normal (Uniprot/RefSeq) and alternative (Splooce) proteins, showing different categories of alternative TIS observed for our data.

Sequences highlighted in orange represent MS peptides found for the Uniprot/RefSeq proteins, and sequences highlighted in yellow represent peptides found exclusively in the alternative sequences from Splooce. Peptides that align specifically to a sequence from Splooce are supposed to characterize ASEs. (A) Alternative TIS is downstream the original one; (B) Same as A, although the beginning of the protein sequence is directly affected by the ASE. (C) Alternative TIS is upstream the original one.

The high proportion of alternative TIS was further explored. All new protein isoforms showing an alternative TIS were searched against the TISdb database (Wan & Qian, 2014), a collection of TIS obtained from a genome-wide method (GTI-Seq) developed by the same authors (Lee et al., 2012). We found that only one TIS present in our list was present in the TISdb providing therefore a proteome validation for that respective TISdb entry. Several reasons could explain the small overlap between the two datasets such as: (i) the different nature of the samples used in both studies, (ii) the fact that most of the TIS present in TISdb are non-canonical and start with others codons than ATG (we restricted our analysis to ATG-associated TIS) and (iii) the lack of proteome validation in most of the studies that populated TISdb.

Wilson and colleagues have suggested that the association between ASE and TIS are restricted to the amino-terminus of proteins where both events are used to produce isoforms that differ at their amino end. Almost 2,000 events like that were identified at the transcriptome level but few (17 instances) were confirmed in a limited search against MS/MS data (Wilson et al., 2014). We wondered whether this type of event would be frequent in our dataset of 355 TIS. Visual inspection of all 355 cases identified only 29 instances (8%) that would fit the model from Wilson et al. (2014) (for more details, see File S2). The low level of validation of such cases at the proteome level, also seen by the authors in their original report, raises doubts about their widespread occurrence. All remaining 326 cases of TIS in our dataset were analyzed to identify the effect of the ASE in the protein sequence originally present in the reference sequence. In only three cases, the alternative TIS was upstream of the original ATG codon. In all remaining cases, the ASE occurred upstream of the alternative TIS and disrupted the respective ORF. An alternative ATG codon, always located downstream of the ASE, is then used as a new TIS. Interestingly, only in 15% of these cases (48 out of 323) the ATG codon used in the TIS is the first one downstream of the ASE.

Conclusions

A limitation one is facing in this type of analysis is the definition of a proper false discovery rate when adding entries in a database ad infinitum. Any observed MS/MS information in such approaches will be tagged to the “best-fit” theoretical peptide present in the database, regardless if that is the correct one. Even though identification engines such as Mascot and MaxQuant (Andromeda) have proof-check algorithms to quantify FDR rate, incorrect MS/MS information might still be reported as true. Therefore there will be always the risk that peptides that are present in the sample but not represented in the database are incorrectly assigned. In addition, there will be a size limit where adding more protein entries created by RNAseq information will be detrimental to the analysis, rather than beneficial. For a good isoform discovery phase study to reliably work, a compromise between database size and validation rounds using complementary databases must be created. A desirable strategy would be to create a collection of public, high quality datasets such as the one used in this work and use them for database-based splicing discovery using different versions of the Splooce database. Recently, similar approaches have been successfully implemented for mapping expressed genes, pseudogenes and characterization of new open reading frames (Kim et al., 2014; Wilhelm et al., 2014), but little was shown regarding splicing isoforms. Therefore, such an approach using Splooce databases with public MS data for ASE discovery is feasible and promising for further characterization of the human proteome draft. Our data demonstrate that by simply complementing routinely used databases with rare/unknown isoform entries predicted by nucleotide sequences approaches, together with already in-use protein identification engines such as MaxQuant/Andromeda can provide satisfactory identification rates without compromising the search engine capabilities.

In summary, a new strategy for the identification of splicing variants in MS/MS data is provided here allowing us to confirm at the proteome level more than 800 new variants. We extended previous observations linking ASE and TIS and provided validation for hundreds of new TIS events. We have upgraded the Splooce portal to take into account the integration of MS/MS data in the validation of splicing variants.

Supplemental Information

File S1 Complete list of peptides identified by MaxQuant using protein entries present in Splooce

Click here for additional data file.

File S2 List and sequence alignment of all identified events characterizing TIS changes

Peptides in yellow are specific to the Splooce database and do not exist in Uniprot as tryptic peptides.

Click here for additional data file.

File S3 List and sequence alignment of all identified events which do not characterize TIS changes

Peptides in yellow are specific to the Splooce database and do not exist in Uniprot as tryptic peptides.

Click here for additional data file.

Abbreviations

ASE Alternative splicing events

TIS Translational initiation site

FDR False discovery rate

GTI-Seq Global translational initiation sequencing

Additional Information and Declarations

Competing Interests

Author Contributions

The authors declare there are no competing interests.

José E. Kroll and Gustavo A. de Souza conceived and designed the experiments, performed the experiments, analyzed the data, contributed reagents/materials/analysis tools, wrote the paper, prepared figures and/or tables, reviewed drafts of the paper.

Sandro J. de Souza conceived and designed the experiments, analyzed the data, contributed reagents/materials/analysis tools, wrote the paper, reviewed drafts of the paper.

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
