# Peer review of "Identification of rare alternative splicing events in MS/MS data reveals a significant fraction of alternative translation initiation sites"

_PeerJ, doi:10.7717/peerj.673_

## Round 0.1 · original submission · Minor Revisions

I feel that the manuscript can be improved by following the suggestions of the reviewers for additional discussion. Please address all suggestions in your response.

·

Basic reporting

Introduction accurately encapsulates the scope, scale and difficulty of the problem with characterisation of alt splice sites using proteomics approaches.
One limitation which is not discussed until much later in the manuscript is the (tremendously variable) quality of product ion spectra (caused by a variety of factors, including analyte concentration and sequence-specific variability in fragmentation events in vacuo). This is important, as poor MS/MS spectra will contribute significantly to the FDR of any identifications: some scoring for the quality of matching made is essential in such an approach.Whilst this point is made later on, it is of sufficient import to make it a matter for the introduction.

Experimental design

One area which could do with significant clarification is the construction of the database used for the MaxQuant (and presumably PEAKS-based) identifications. Were all entries from UniProt Human, with RefSeq-based alt splice data supplementing? This needs to be made much clearer. A simple Venn diagram illustrating the sequences used in each db search applied (with numbers of sequences) would clarify this significantly. Some mention of how the entries were tagged (as Fig 1 clearly indicates that they could be pulled out readily) would be valuable. As a further point, it is customary to indicate download date, source and number of sequences for sequence databases where downloaded from common sources (e.g. UniProt Human, downloaded from [url] date 13/12/13, 89,000 entries). I would also be careful about terminology here – calling the database used for searching a “dataset” as it is stated in the text risks confusion with the MS dataset used for the searches.

Validity of the findings

The box & whisker plots in Fig 2 also need more explanation: the points on the plots extend significantly beyond the ‘whiskers’ (the iqr on a conventional plot), both for intensity and ion scores, so this is not a conventional plot – this needs explanation. How were these generated? Is this part of MaxQuant or an independent analysis of the MaxQuant results?
In table 1 only 463 of the ASEs are categorised: where are the other ASEs? Are these something to do with the alternative TIS mentioned in the next section (although this still doesn’t add up to the 892)? The numbers are confusing in this section – there is room for significantly more clarity here.

Additional comments

Overall this represents an interesting and valuable study to identify ASEs in large proteomics datasets. These are phenomena which are currently ignored by most large-scale omics studies, therefore it is important that we find ways to identify these species where they are present in data.
It would be valuable if comment on the number of events (and hence to the number of unidentified/misidentified MS/MS spectra in such datasets) could be made - this is valuable information for proteomics researchers. Comment on how the authors' strategy could contribute to database searching/algorithm development in the future would be valuable.

·

Basic reporting

There are a few typos but other than that the article is well written with sufficient information provided.

Experimental design

Was trypsin with restriction of proline used? (Lines 95-107)

Lines 120-142: It would be interesting to see what the results would have been if you had searched the Splooce database only, obviously the number of identified peptides would decrease. With the smaller search space there may be some additional peptides which were filtered out in the combined database search. Though the article does not require this for submission it may produce some interesting findings.

Validity of the findings

The findings from the MS data match those observed in the transcriptomics validating the results.

Additional comments

This approach of searching a splice variant database opens up a new set of peptides to identify proteins from MS datasets. As proteomic experiments mine deeper and deeper into the proteomes these type of databases will become increasingly important. Currently in most MS/MS analyses there are significant number of unassigned spectra, this offers an explanation for some of these events.

There are a few typos but overall this is a very well written and presented piece of research.

There is a typo on line 113 Carbamydomethyl should be carbomamidomethyl

line 151: change "such" to "the"

Line 178: Change "spam" to "span)

Reviewer 3 ·

Basic reporting

No Comments

Experimental design

No Comments

Validity of the findings

No Comments

Additional comments

This manuscript introduce used of Splooce, a database of splicing variants recently developed by the authors, to search MS data derived from a variety of human tumor cell lines. More than 800 new protein variants were identified whose corresponding MS spectra were specific to protein entries from Splooce. The manuscript is well-structured, but still some minor comments and suggestions should be addressed before the publication:

Minor comments:
1. Some references are missing:
1.1 - For example about the complexity of databases it would be good if the author use previous studies of proteomics databases such as PMID: 21658481 and PMID: 23802565
1.2 -The manuscript entitle "Consequences of the discontinuation of the International Protein Index (IPI) database and its substitution by the UniProtKB "complete proteome”
sets" discussed the consequences of IPI discontinuation that the authors mentioned in line PMID: 21932440.
1.3 - You can use this reference for an update of proteomics resources and databases line 40: PMID: 25158685.

2. It would be interesting to see a workflow about how to generate the databases using the Splooce platform. I know the manuscript is about the use of the tool, but it is also a way to highlight and explain in details the current workflow followed by the authors.

3. One point missing in the discussion is about the combination of the search engines results. It would be great to have some discussion about how to combine the results from MaxQuant and PEAKS taking to account error propagation at peptide level and protein level.

4. The are some other tools to generate databases from RNA-seq evidences, It would be interesting to see in the manuscript one section devoted to to the use of those tools (SpliceDB or customProDB) and see how the many evidences the authors can get from them.

---

## Round 0.2 · accepted · Accept

Thank you for returning your comments. I am pleased that we are able to accept this version of the manuscript.